# LEARNING TO LEARN
# WITH CONDITIONAL CLASS DEPENDENCIES

**Xiang Jiang**[*†]**, Mohammad Havaei**[*]**, Farshid Varno**[*†]**, Gabriel Chartrand**[*]**,
Nicolas Chapados**[*]**, Stan Matwin**[†]

[*]Imagia Inc., [†]Dalhousie University
{xiang.jiang,mohammad,farshid.varno,gabriel,nic}@imagia.com, stan@cs.dal.ca

## ABSTRACT

Neural networks can learn to extract statistical properties from data, but they seldom make use of structured information from the label space to help representation learning. For example "cat" and "dog" are closer than "cat" and "truck". Although some label structure can implicitly be obtained when training on huge amounts of data, in a few-shot learning context where little data is available, making explicit use of the label structure can inform the model to reshape the representation space to reflect a global sense of class dependencies. We propose a meta-learning framework, Conditional class-Aware Meta-Learning (CAML), that conditionally transforms feature representations based on a metric space that is trained to capture inter-class dependencies. This enables a conditional modulation of the feature representations of the base-learner to impose regularities informed by the label space. Experiments show that the conditional transformation in CAML leads to more disentangled representations and achieves competitive results on the *mini*ImageNet benchmark.

## 1 INTRODUCTION

In machine learning, the objective of classification is to train a model to categorize inputs into various classes. We usually assume a categorical distribution over the label space, and thus effectively ignore dependencies among them. However, class structure does exist in real world and is also present in most datasets. Although class structure can be implicitly obtained as a by-product during learning, it is not commonly exploited in an explicit manner to develop better learning systems. The use of label structure might not be of prime importance when having access to huge amounts of data, such the full ImageNet dataset. However, in the case of few-shot learning where little data is available, meta-information such as dependencies in the label space can be crucial.

In recent years, few-shot learning—learning from few examples across many tasks—has received considerable attention (Ravi & Larochelle, 2016; Snell et al., 2017; Finn et al., 2017; Vinyals et al., 2016). In particular, the concept of meta-learning has been shown to provide effective tools for few-shot learning tasks. In contrast to common transfer learning methods that aim to fine-tune a pre-trained model, meta-learning systems are trained by being exposed to a large number of tasks and evaluated in their ability to learn new tasks effectively. In meta-training, learning happens at two levels: a meta-learner that learns across many tasks, and a base-learner that optimizes for each task. Model-Agnostic Meta-Learning (MAML) is a gradient-based meta-learning algorithm that provides a mechanism for rapid adaptation by optimizing only for the initial parameters of the base-learner (Finn et al., 2017).

Our motivation stems from a core challenge in gradient-based meta-learning, wherein the quality of gradient information is key to fast generalization: it is known that gradient-based optimization fails to converge adequately when trained from only a few examples (Ravi & Larochelle, 2016), hampering the effectiveness of gradient-based meta-learning techniques. We hypothesize that under such circumstances, introducing a metric space trained to encode regularities of the label structure can impose global class dependencies on the model. This class structure can then provide a high-level view of the input examples, in turn leading to learning more disentangled representations.

We propose a meta-learning framework taking advantage of this class structure information, which is available in a number of applications. The Conditional class-Aware Meta-Learning (CAML) model is

tasked with producing activations in a manner similar to a standard neural network, but with the additional flexibility to shift and scale those activations conditioned on some auxiliary meta-information. While there are no restrictions on the nature of the conditioning factor, in this work we model class dependencies by means of a metric space. We aim to learn a function mapping inputs to a metric space where semantic distances between instances follow an Euclidean geometry—classes that are semantically close lie in close proximity in an $\ell^p$ sense. The goal of the conditional class-aware transformation is to make explicit use of the label structure to inform the model to reshape the representation landscape in a manner that incorporates a global sense of class structure.

The contributions of this work are threefold: (i) We provide a meta-learning framework that makes use of structured class information in the form of a metric space to modulate representations in few-shot learning tasks; (ii) We introduce class-aware grouping to improve the statistical strength of few-shot learning tasks; (iii) We show experimentally that our proposed algorithm learns more disentangled representation and achieves competitive results on the *mini*ImageNet benchmark.

## 2 BACKGROUND

We start by describing the meta-learning formulation proposed by Vinyals et al. (2016) and Ravi & Larochelle (2016), and review MAML (Finn et al., 2017), of which CAML is an instance.

### 2.1 META-LEARNING PROBLEM FORMULATION

The goal of meta-learning is to learn from a *distribution of tasks*. The learning happens on two levels: (i) a meta-level model, or meta-learner, that learns across many tasks, and (ii) a base-level model, or base-learner, that operates within each specific task. Meta-learning happens in task space, where each task can be treated as one meta-example. In the meta-learning formulation, we define a collection of regular tasks as meta-sets $\mathscr{D}$, and each task $\mathcal{D} \in \mathscr{D}$ has its own $\mathcal{D}^{\text{train}}$ and $\mathcal{D}^{\text{test}}$ split. $\mathcal{D}^{\text{train}}$ is often denoted as the "support set" and $\mathcal{D}^{\text{test}}$ the "query set". The resulting meta-learner objective is to choose parameters $\theta$ that minimize the expected loss $\mathcal{L}(\cdot;\theta)$ across all tasks in $\mathscr{D}$,

$$\theta^* = \operatorname{argmin}_\theta \mathbb{E}_{\mathcal{D} \sim \mathscr{D}}[\mathcal{L}(\mathcal{D};\theta)].$$

At the meta-level, the meta-sets $\mathscr{D}$ can be further split into disjoint meta-training set $\mathscr{D}_{\text{meta-train}}$, meta-validation set $\mathscr{D}_{\text{meta-valid}}$ and meta-test set $\mathscr{D}_{\text{meta-test}}$. The meta-learner is trained on $\mathscr{D}_{\text{meta-train}}$, validated on $\mathscr{D}_{\text{meta-valid}}$ and finally evaluated on $\mathscr{D}_{\text{meta-test}}$.

### 2.2 MODEL-AGNOSTIC META-LEARNING

Model-Agnostic Meta-Learning (Finn et al., 2017) is a meta-learning algorithm that aims to learn representations that encourage fast adaptation across different tasks. The meta-learner and base-learner share the same network structure, and the parameters learned by the meta-learner are used to initialize the base-learner on a new task.

To optimize the meta-learner, we first sample a set of tasks $\{\mathcal{D}_1, \mathcal{D}_2, ..., \mathcal{D}_S\}$ from the meta-training set $\mathscr{D}_{\text{meta-train}}$. For a meta-learner parameterized by $\theta$, we compute its adapted parameters $\theta_i$ for each sampled task $\mathcal{D}_i$. The adapted parameters $\theta_i$ are task-specific and tell us the effectiveness of $\theta$ as to whether it can achieve generalization through one or a few additional gradient steps. The objective of the meta-learner is to optimize the representation $\theta$ such that it leads to good task-specific adaptations $\theta_i$ with only a few gradient steps. The meta-learner performs slow learning at the meta-level across many tasks to support fast learning on new tasks. At meta-test time, we initialize the base-learner with the meta-learned representation $\theta^*$ followed by gradient-based fine-tuning.

## 3 METHOD

### 3.1 CONDITIONAL CLASS-AWARE META-LEARNING

As shown in Figure 1, the proposed Conditional class-Aware Meta-Learning (CAML) is composed of four components: an embedding function $f_\phi$ that maps inputs to a metric space, a base-learner $f_\theta$ that learns each individual task, an adaptation function $f_c$ that conditionally modulates the representations of

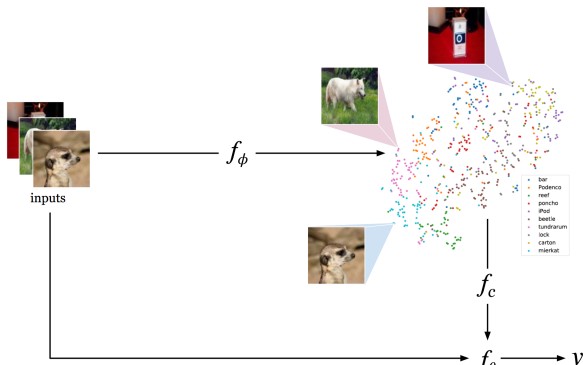

Figure 1: Overview of Conditional class-Aware Meta-Learning. Inputs to the model are mapped onto an embedding space using $f_\phi$ which are then used to modulate the base-learner $f_\theta$ through a conditional transformation $f_c$. We use MAML (not shown) to meta-learn $f_c$, $f_\theta$, and a metric loss to pre-train $f_\phi$

the base-learner, and a meta-learner that learns across different tasks. Figure 1 depicts a toy illustration of the task inference procedure where examples from three classes are mapped onto a metric space using $f_\phi$, which are further used to modulate the base-learner $f_\theta$ through a conditional transformation function $f_c$.

The main contribution of this paper is to incorporate metric-based conditional transformations ($f_c$) into the meta-learning framework at the instance level. A notable feature of the proposed method is that the model has a global sense of the label space through the embedding function $f_\phi$ by mapping examples onto the semantically meaningful metric space. The embeddings on the metric space inform the base-learner $f_\theta$ about the label structure which in turn helps disentangle representations from different classes. This structured information can also provide a global view of the input examples to improve gradient-based meta-learning.

In a simplistic form, our proposed model makes predictions using

$$\hat{y} = f_\theta\Big(x; f_c\big(f_\phi(x)\big)\Big),$$

where the base-learner $f_\theta$ is conditioned on the embedding space $f_\phi(x)$ through the conditional transformation $f_c$. This is in contrast with a regular base-learner where $\hat{y} = f_\theta(x)$. In our framework, we use MAML to meta-learn $f_c$ and $f_\theta$. The metric space is pre-trained using distance-based loss function.

## 3.2 METRIC SPACE AS CONDITIONAL INFORMATION

We encode information of the label structure through $f_\phi$ in the form of an $M$-dimensional metric space, where each input example is reduced to a point in the metric space. The goal of the metric learning step is to optimize parameter $\phi$ such that distances between examples in the metric space are semantically meaningful. Given the parameters of the metric space $\phi$, which is represented by a convolutional network, we calculate a centroid $\mathbf{c}_t$ for each class $t$,

$$\mathbf{c}_t = \frac{1}{K} \sum_{(\mathbf{x}_i, y_i) \in \mathcal{D}^{\text{train}}} \mathbb{1}_{\{y_i = t\}} f_\phi(\mathbf{x}_i),$$

where $K$ denotes the number of examples for class $t$, $\mathbb{1}_{\{y_i=t\}}$ denotes an indicator function of $y_i$ which takes value 1 when $y_i = t$ and 0 otherwise. The centroid $\mathbf{c}_t$ is the sample mean among all instances from the same class which is treated as a prototype representation of the class $t$. The mapping function $f_\phi$ is optimized to minimize the negative log-probability defined in Eq. (1) by minimizing the Euclidean distance $d$ between an example and its corresponding class centroid $\mathbf{c}_t$ while maximizing its Euclidean distance to other class centroids $\mathbf{c}_{t'}$:

$$\underset{\phi}{\operatorname{argmin}} \mathbb{E}\left[ d(f_\phi(\mathbf{x}_i), \mathbf{c}_t)) + \log \sum_{t'} \exp(-d(f_\phi(\mathbf{x}_i), \mathbf{c}_{t'})) \right]. \tag{1}$$

In relation to prototypical networks (Snell et al., 2017), we use the same loss function for metric learning. However, these frameworks differ in the test mode: we are not interested in example-centroid distances

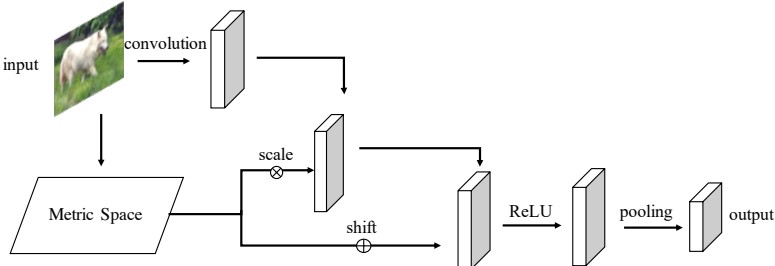

Figure 2: Conditionally transformed convolutional block. The convolutional feature maps are conditionally scaled and shifted based on the input image's representation in the pre-trained metric space.

for label assignment, but rather in the projection $f_\phi(\mathbf{x}_i)$ from the input space to the metric space that encapsulates inferred class regularities given the input example $\mathbf{x}_i$. In relation to other pre-training methods that use the meta-train classes to train a 64-way classifier, our use of the metric space imposes distance-based constraints to learn embeddings that follow semantically meaningful distance measures.

We empirically find it difficult to optimize both the metric space and base-learner end-to-end. The metric space is pre-trained on the meta-train data and it is not updated during meta-learning. This also ensures the metric space is trained on a large number of classes to capture the global class dependencies.

## 3.3 CONDITIONALLY TRANSFORMED CONVOLUTIONAL BLOCK

We now turn to describing the conditionally transformed convolutional block, shown in Figure 2, which uses the metric space described in Section 3.2 to inform the base-learner about the label structure of a task. The conditional transformation $f_c$ receives embeddings from the metric space and produces transformation operations to modulate convolutional representations of the base-learner $f_\theta$.

Our conditional transformation has close relation to Batch Normalization (BN) (Ioffe & Szegedy, 2015) that normalizes the input to every layer of a neural network. In order to conditionally modulate feature representations, we use Conditional Batch Normalization (CBN) (Dumoulin et al., 2017) to predict scale and shift operators from conditional input $\mathbf{s}_i$:

$$\hat{\gamma}_c = f_{c,\gamma}(\mathbf{s}_i), \qquad \hat{\beta}_c = f_{c,\beta}(\mathbf{s}_i), \tag{2}$$

where $f_{c,\gamma}$ and $f_{c,\beta}$ can be any differentiable function. This gives our model the flexibility to shift or scale the intermediate representations based on some source information in $\mathbf{s}_i$. Since examples belonging to the same class are conceptually close, we exploit this inherent relationship in the metric space to modulate the feature maps at the example level in a way that encodes the label structure.

Once we obtained the embedding function $f_\phi$, we use two auxiliary networks, learned end-to-end together with the meta-learner, to predict the shift and scale factors of the convolutional feature map:

$$\hat{\gamma}_{i,c} = f_{c,\gamma}(f_\phi(\mathbf{x}_i)), \qquad \hat{\beta}_{i,c} = f_{c,\beta}(f_\phi(\mathbf{x}_i)). \tag{3}$$

Having computed $\hat{\gamma}_{i,c}$ and $\hat{\beta}_{i,c}$, Conditional Batch Normalization (CBN) is applied as follows:

$$\text{CBN}(\mathbf{R}_{i,c}|\hat{\gamma}_{i,c},\hat{\beta}_{i,c}) = \hat{\gamma}_{i,c}\frac{\mathbf{R}_{i,c} - \mathbb{E}[\mathbf{R}_c]}{\sqrt{\text{Var}[\mathbf{R}_c] + \epsilon}} + \hat{\beta}_{i,c}, \tag{4}$$

where $\mathbf{R}_{i,c}$ refers to the $c^{th}$ feature map from the $i^{th}$ example, $\epsilon$ is a small constant, $\beta_c$ and $\gamma_c$ are learnable parameters shared within a task. $\mathbb{E}[\mathbf{R}_c]$ and $\text{Var}[\mathbf{R}_c]$ are batch mean and variance of $\mathbf{R}_c$.

It is worthwhile to note the effect of conditional transformation. The conditional bias transformation with $\hat{\beta}_{i,c}$ is analogous to concatenation-based conditioning where the conditional information is concatenated to the feature maps (Dumoulin et al., 2018). The conditional scaling factor provides multiplicative interactions between the metric space and the feature maps to aggregate information.

Furthermore, the goal of the conditionally transformed convolutional block is to simultaneously capture the two views of a classification task: a global view that is aware of the relationships among all classes, and a local views of the current N-way K-shot classification task. The metric space, or the global view,

is pre-trained in a way that is independent of the current N-way K-shot task; while the base-learner, or the local view, attempts to develop representations for the current classification task. Although the metric space is never trained on the meta-test classes, we expect the learned metric space to generalize from the meta-train tasks to the meta-test tasks.

We further describe parameter sharing for CBN learning in Section 3.3.1, and class-aware grouping in Section 3.3.2 which provides more statistical strength for more effective few-shot learning.

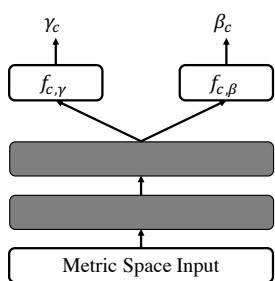

Figure 3: CBN shared architecture

Figure 4: Normalization methods. 'C' denotes channels, 'H, W' spatial dimensions and 'N' examples.

### 3.3.1 MULTITASK LEARNING OF CBN

Although one can predict $\hat{\gamma}_c$ and $\hat{\beta}_c$ using two separate functions, we find it beneficial to use shared parameters as shown in Figure 3. The shared representations are more efficient at producing conditional transformations which also provide a strong inductive bias to help learning (Caruana, 1997).

### 3.3.2 CLASS-AWARE GROUPING

We propose class-aware grouping, as shown in Figure 4 (b), to further exploit properties of metric space. The motivation stems from a lack of statistical strength when learning from only a few examples. As an example, in $N$-way 1-shot learning, the model is required to find the most meaningful way to distinguish different classes. However, gradient-based optimization may lead to the discovery of irrelevant features which coincide with the class labels, such as background colors.

We address this problem by class-aware grouping that is guided by our metric space. This is related to "transduction", which is a standard technique in MAML-based methods. Transduction as discussed in Nichol et al. (2018), makes use of the channel mean $\mathbb{E}[\mathbf{R}_c]$ and variance $\text{Var}[\mathbf{R}_c]$, defined in Eq. (4), of query examples when evaluating a base-learner. In contrast to standard transduction methods that calculate mean and variance over all examples of the current batch, we introduce class-aware grouping that clusters examples into different groups and use group-based mean and variance to normalize different channels. The grouping is determined by distance measures in the metric space where examples are grouped together based on their nearest centroid $c_t$ defined in Section 3.2. Class-aware grouping is integrated into CBN as:

$$\text{CBN}(\mathbf{R}_{i,c}|\hat{\gamma}_{i,c},\hat{\beta}_{i,c}) = \hat{\gamma}_{i,c}\frac{\mathbf{R}_{i,c} - \mathbb{E}[\mathbf{R}_{i,c} \cdot \mathbb{1}_{\{x_i \in t\}}]}{\sqrt{\text{Var}[\mathbf{R}_{i,c} \cdot \mathbb{1}_{\{x_i \in t\}}] + \epsilon}} + \hat{\beta}_{i,c}, \tag{5}$$

where $\mathbb{1}_{\{x_i \in t\}}$ indicates if an example $x_i$ belongs to cluster $t$, and $\mathbb{E}[\mathbf{R}_{i,c} \cdot \mathbb{1}_{\{x_i \in t\}}]$ represents the average of channel $\mathbf{R}_c$ among examples clustered at $c_t$. This is depicted in Figure 4 where the channel mean and variance are calculated for every group.This approach informs the base-learner about what to expect from the query examples at the class level through channel mean and variance, which provides more explicit guidance to the meta-learning procedure.

### 3.4 TRAINING DETAILS

The base-learner ($f_\theta$) is composed of 4 layers of $3 \times 3$ convolutions with a $4 \times 4$ skip connections from the input to the final convolutional layer. The use of skip connections is to improve the gradient flow as MAML unfolds the inner loop into one computational graph. The use of skip connections is empirically important to the proposed model. Each convolutional layer has 30 channels and is followed by CBN, ReLU and $2 \times 2$

max-pooling operations. The output of the final convolution is flattened and fed to a 1-layer dense classifier. For learning the metric space ($f_\phi$), we use the same residual network (ResNet-12) as Oreshkin et al. (2018). The metric space is pre-trained on the same meta-training dataset for 30,000 episodes and not updated while learning the base-learner. The meta-learner is trained for 50,000 episodes. We empirically observe that training the metric space and meta-learner end-to-end is overly complex and prone to over-fitting. For CBN functions ($f_c$), we use 3 dense layers with 30 hidden units each. Every layer is followed by a ReLU except for the last layer where no activation is used. For the meta-learner, we use MAML with 1 gradient step for 1-shot learning and 5 gradient steps for 5-shot learning. We use the Adam (Kingma & Ba, 2014) optimizer and clip the L2 norm of gradients with an upper bound of 5. Similar to MAML, we use transduction where the statistics of the current batch is used for $\mathbb{E}(.)$ and Var$(.)$ in Eq. (4) for both training and testing.

## 4 RELATED WORK

### 4.1 META-LEARNING

Meta-learning or "learning-to-learn" (Schmidhuber, 1987; Bengio et al., 1992; Mitchell & Thrun, 1993; Vilalta & Drissi, 2002) has been studied as a means to acquire meta-knowledge across many tasks. In recent years, meta-learning has become an important approach for few-shot learning. A number of approaches aim to learn universal learning procedure approximators by supplying training examples to the meta-learner that outputs predictions on testing examples (Hochreiter et al., 2001; Vinyals et al., 2016; Santoro et al., 2016; Mishra et al., 2017). Other approaches learn to generate model parameters conditioned on training examples (Gomez & Schmidhuber, 2005; Munkhdalai & Yu, 2017; Ha et al., 2016; Gidaris & Komodakis, 2018), or learning optimization algorithms across different tasks (Ravi & Larochelle, 2016; Andrychowicz et al., 2016; Li & Malik, 2017).

#### 4.1.1 GRADIENT-BASED META-LEARNING

Our work is more inline with gradient-based meta-learning that aims to learn representations that encourage fast adaptation on new tasks. These methods are based on model-agnostic meta-learning (MAML) introduced by Finn et al. (2017). While the original MAML requires second-order gradients in meta-optimization, REPTILE (Nichol et al., 2018) only uses first-order gradient information. Furthermore, Latent Embedding Optimization (LEO) (Rusu et al., 2018) is proposed to perform gradient-based optimization on a low-dimensional latent space instead of the original high-dimensional parameter space. We emphasize that all those methods do not make explicit use of structured label information, which is a main novelty in this paper.

#### 4.1.2 METRIC-BASED META-LEARNING

Our work also relates closely to metric-based meta-learning that learns a metric space across different tasks. Siamese networks (Koch et al., 2015) learn a similarity measure between inputs using a shared network architecture that outputs high probability when paired examples are from the same class. Matching networks (Vinyals et al., 2016) use full context embeddings to encode examples to the metric space and use attention as a similarity measure for predictions. Prototypical networks (Snell et al., 2017) compute a centroid, or prototype, for every class that are later used for distance-based queries of new examples. Task dependent adaptive metric (TADAM) (Oreshkin et al., 2018) uses metric scaling based on tasks representations to learn a task-dependent metric space.

A notable difference between the metric-based methods and our approach is that, the metric space in our model is not aimed for distance-based classification. Rather, we use the metric space to represent class structure which facilitates the gradient-based meta learning towards better generalization. Another difference between our method and TADAM is that, TADAM scales the metric space at the task level where all examples within a task are scaled in the same manner. In contrast, our method provides instance-based conditioning that makes use of the precise representation of each example. Put another way, TADAM modulates the inference on a metric space from a task perspective, while CAML uses example-level representation to modulate the representation at the content level.

### 4.2 CONDITIONAL TRANSFORMATION

In style transfer, conditional instance normalization is proposed by Dumoulin et al. (2017) that transforms the content image conditioned on the domain of the style image. In visual question answering, De Vries

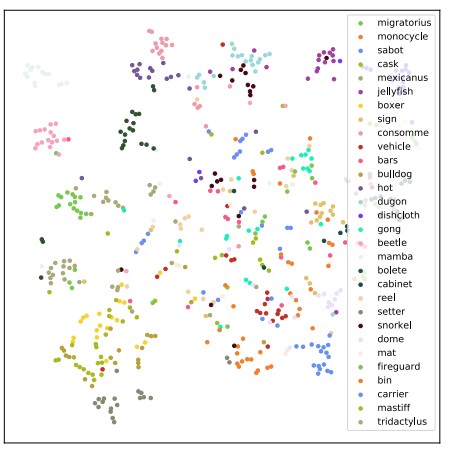
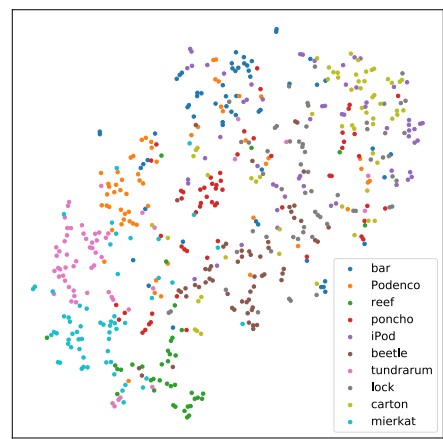

(a): Randomly selected 30 training labels          (b): Randomly selected 10 validation labels

Figure 5: t-SNE visualization of the learned metric space colored by category.

et al. (2017) have shown that it is beneficial to modulate early visual signals of a pre-trained residual network by language in the form of conditional batch normalization. It was further shown that feature-wise linear modulation (Perez et al., 2017; Dumoulin et al., 2018) can efficiently select meaningful representations for visual reasoning.

The notion that is common to all these methods is the use of an additional input source, e.g., style or language, to conditionally transform intermediate representations of a network. In few-shot learning, Zhou et al. (2018) suggested that it is easier to operate in the concept space in the form of a lower dimensional representation. This is compatible with our proposed approach that uses the metric space as concept-level representation to modulate intermediate features of the base-learner.

## 5  EXPERIMENTS

We use *mini*ImageNet to evaluate the proposed Conditional class-Aware Meta-Learning algorithm. *mini*ImageNet (Vinyals et al., 2016) is composed of 84×84 colored images from 100 classes, with 600 examples in each class. We adopt the class split by Ravi & Larochelle (2016) that uses 64 classes for training, 16 for validation, and 20 for test. For $N$-way $K$-shot training, we randomly sample $N$ classes from the meta-train classes each containing $K$ examples for training and 20 examples for testing. At meta-testing time, we randomly sample 600 $N$-way $K$-shot tasks from the test classes.

### 5.1  RESULTS

The results presented in Table 1 show that our proposed algorithm has comparable performance on the state-of-the-art *mini*ImageNet 5-way 1-shot classification task, and competitive results on the 5-way 5-shot task. Unlike LEO (Rusu et al., 2018) that applies meta-learning on pre-trained representations, our meta-learner is able to effectively operate on the high-dimensional parameter space. Our method also does not require co-training compared with TADAM (Oreshkin et al., 2018).

Figure 5 shows the t-SNE plot of the learned metric space for both meta-train and meta-validation classes. As seen in Figure 4b, examples from the meta-validation set form clusters consistent with their class membership, even though the metric space is not trained on these classes. For example, "mierkat", "tundrarum" and "podenco" are all animals and they are clustered close together.

The first main baseline we report is MAML. CAML improves upon MAML by about 10% on both 1-shot and 5-shot tasks. This means incorporating class dependencies in the form of a metric space can greatly facilitate gradient-based meta-learning. We also compare with MAML using our base-learner architecture equipped with skip connections from the input to the last convolutional layer. MAML trained with our base-learner's architecture yields similar performance as the original MAML, suggesting the improvement is resulted from the proposed CAML framework, rather than changes in the base-learner's architecture.

Table 1: *mini*ImageNet classification accuracy with 95% confidence intervals.

| Model | 5-way 1-shot | 5-way 5-shot |
|---|---|---|
| Meta-Learner LSTM (Ravi & Larochelle, 2016) | 43.44% ± 0.77% | 60.60% ± 0.71% |
| Matching Networks (Vinyals et al., 2016) | 46.6% | 60.0% |
| Prototypical Network with Soft k-Means (Ren et al., 2018) | 50.41% ± 0.31% | 69.88% ± 0.20% |
| MetaNet (Munkhdalai & Yu, 2017) | 49.21% ± 0.96% | − |
| TCML (Mishra et al., 2018) | 55.71% ± 0.99% | 68.88% ± 0.92% |
| adaResNet (Munkhdalai et al., 2018) | 56.88% ± 0.62% | 71.94 ± 0.57% |
| Cosine Classifier (Gidaris & Komodakis, 2018) | 56.20% ± 0.86% | 73.00% ± 0.64% |
| TADAM (Oreshkin et al., 2018) | 58.5% | 76.7% |
| LEO (Rusu et al., 2018) | **61.76% ± 0.08%** | **77.59% ± 0.12%** |
| MAML (Finn et al., 2017) | 48.7% ± 1.84% | 63.11% ± 0.92% |
| MAML on our architecture | 48.26% ± 1.04% | 64.25% ± 0.78% |
| Prototypical Network (Snell et al., 2017) | 49.42% ± 0.78% | 68.2% ± 0.66% |
| Prototypical Network on our metric space | 55.96% ± 0.91% | 71.64% ± 0.70% |
| CAML (with multitask learning alone) | 52.56% ± 0.83% | 71.35% ± 1.13% |
| CAML (with class-aware grouping alone) | 55.28% ± 0.90% | 71.14% ± 0.81% |
| CAML (full model) | 59.23% ± 0.99% | 72.35% ± 0.71% |

The confidence intervals are constructed by sampling 600 evaluation tasks from the meta-test classes.

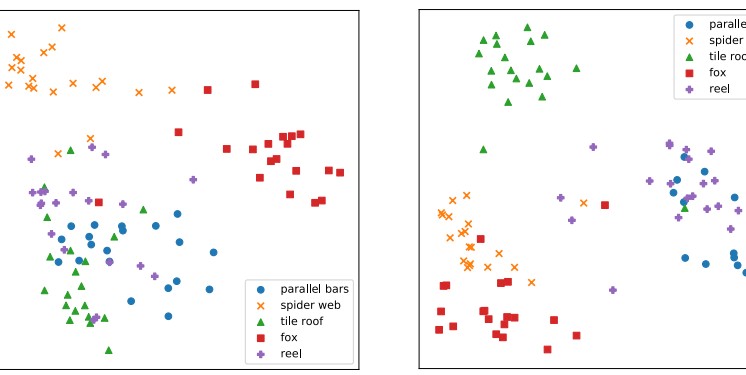

(a): Before conditional transformation    (b): After conditional transformation

Figure 6: PCA visualization of feature maps from the last convolutional layer colored by category.

The second baseline we use is prototypical network. We measure the classification ability of our metric space using prototypical network as a classifier, shown in Table 1 (Prototypical Network in our metric space). These results suggest that making predictions on the metric space alone is inferior to CAML.This can be explained by CAML's ability to fast-adapt representations even when the metric space does not provide good separations. We also find that CAML has larger improvements in 1-shot tasks than 5-shot ones. This is because, in 1-shot learning, metric-based methods estimate class representations from a single example, making it difficult to provide a robust class estimation.

## 5.2 THE EFFECT OF CONDITIONAL TRANSFORMATION

We compare activations before and after the conditional transformation to better understand how conditional transformation modulates the feature representations. Figure 6 shows the PCA projections of the last convolutional layer in the base-learner. We observe in Figure 5a that, before conditional transformation, examples from three classes ("parallel bars", "tile roof" and "reel") are mixed together. In Figure 5b, after the conditional transformation is applied, one of the previously cluttered classes ("tile roof") become separated from the rest classes. This confirms that metric space can alleviate the difficulty in few-shot learning by means of conditional transformations.

We undertake ablation studies to show the impact of multitask learning and class-aware grouping. Empirical results in Table 1 suggest that, while 1-shot learning is sensitive to multitask learning and class-aware grouping, 5-shot learning is not affected by those techniques. This is owing to a lack of statistical strength in 1-shot learning, which requires more explicit guidance in the training procedure. This means exploiting metric-based channel mean and variance can provide valuable information to improve meta-learning. More detailed ablation studies are included in Appendix A.

## 6  CONCLUSION

In this work, we propose Conditional class-Aware Meta-Learning (CAML) that incorporates class information by means of an embedding space to conditionally modulate representations of the base-learner. By conditionally transforming the intermediate representations of the base-learner, our goal is to reshape the representation with a global sense of class structure. Experiments reveal that the proposed conditional transformation can modulate the convolutional feature maps towards a more disentangled representation. We also introduce class-aware grouping to address a lack of statistical strength in few-shot learning. The proposed approach obtains competitive results with the current state-of-the-art performance on 5-way 1-shot and 5-shot *mini*ImageNet benchmark.

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

## A    ADDITIONAL ABLATION STUDIES

### A.1    THE IMPACT OF MULTITASK LEARNING AND CLASS-AWARE GROUPING

To better understand the role of different components in the proposed conditional transformation, we undertake ablation studies to provide further insights into CBN. We study the impact of multitask learning detailed in Section 3.3.1 and class-aware grouping described in Section 3.3.2.

Table 2: Ablation study on the impact of multitask learning and class-aware grouping *mini*ImageNet.

| CBN | multitask | class-aware grouping | 5-way 1-shot | 5-way 5-shot |
|-----|-----------|----------------------|--------------|--------------|
| × | × | × | $48.26\% \pm 1.04\%$ | $64.25\% \pm 0.78\%$ |
| ✓ | × | × | $52.06\% \pm 1.12\%$ | $69.84\% \pm 1.28\%$ |
| ✓ | ✓ | × | $52.56\% \pm 0.83\%$ | $71.35\% \pm 1.13\%$ |
| ✓ | × | ✓ | $55.28\% \pm 0.90\%$ | $71.14\% \pm 0.81\%$ |
| ✓ | ✓ | ✓ | $59.23\% \pm 0.99\%$ | $72.35\% \pm 0.71\%$ |

Table 3: Ablation study on the impact of conditional transformation operators for *mini*ImageNet.

| Model | BN | CBN with $\hat{\beta}_c$ alone | CBN with $\hat{\gamma}_c$ alone | CBN |
|-------|-----|------------------------------|------------------------------|-----|
| 5-way 1-shot | $48.7\% \pm 1.84\%$ | $56.04\% \pm 0.99\%$ | $\mathbf{57.83\% \pm 1.04\%}$ | $\mathbf{59.23\% \pm 0.99\%}$ |

Empirical results from Table 2 suggest that, while 1-shot learning is sensitive to multitask learning and class-aware grouping, 5-shot learning is less sensitive those techniques. This is owing to a lack of sufficient training examples in 1-shot learning tasks, which requires more explicit guidance in the training procedure. We further note that, in 1-shot learning, using class-aware grouping alone can improve CBN's performance by 3%. This means exploiting metric-based channel mean and variance can provide valuable information for gradient-based meta-learning.

## A.2    THE IMPACT OF SCALE AND SHIFT TRANSFORMATIONS

For CBN parameters, we observe that more than half of the predicted $\hat{\beta}_c$ are negative. This is inline with findings from Perez et al. (2017) that CBN selectively suppresses activations of a feature map when followed by a ReLU. To further examine the impact of the scale and shift operators, we train CBN with each operator alone. Table 3 shows CBN works the best when both $\hat{\gamma}_c$ and $\hat{\beta}_c$ are used, and $\hat{\gamma}_c$ contributes more than $\hat{\beta}_c$, owing to its multiplicative interactions between the metric space and convolutional feature representations.

