# OpenReview forum: "Learning to Learn with Conditional Class Dependencies"
_ICLR.cc/2019/Conference_

### Official Review · AnonReviewer2 · 2018-10-16
**A paper with clear idea for few-shot learning, but there are still some questions about the paper.**

**Rating:** 4
**Confidence:** 5

**Review:**

This paper proposes a new few-shot learning method with class dependencies. To consider the structure in the label space, the authors propose to use conditional batch normalization to help change the embedding based on class-wise statistics. Based on which the final classifier can be learned by the gradient-based meta-learning method, i.e., MAML. Experiments on MiniImageNet show the proposed method can achieve high-performance, and the proposed part can be proved to be effective based on the ablation study.

There are three main concerns about this paper, and the final rating depends on the authors' response.
1. The motivation
The authors claim the label structure is helpful in the few-shot learning. If the reviewer understands correctly, it is the change of embedding network based on class statistics that consider such a label structure. From the objective perspective, there are no terms related to this purpose, and the embedding space learning is also based on the same few-shot objective. Will it introduces more information w.r.t. only using embedding space to do the classification?

2. The novelty.
This paper looks like a MAML version of TADAM. Both of the methods use the conditional batch normalization in the embedding network, while CAML uses MAML to learn another classifier based on the embedding. Although CAML uses the CBN at the example level and considers the class information in a transductive setting, it is not very novel. From the results, the proposed method uses a stronger network but does not improve a lot w.r.t. TADAM.

3. Method details
3.1 Since CBN is example induced, will it prone to overfitting?
3.2 About the model architecture.
CAML uses a 4*4 skip connection from input to output. It is OK to use this improve the final performance, but the authors also need to show the results without the skip connection to fairly compare with other methods. Is this skip connection very important for this particular model? Most methods use 64 channel in the convNet while 30 channels are used in this paper. Is this computational consideration or to avoid overfitting? It is a bit strange that the main network is just four layers but the conditional network is a larger and stronger resNet.
3.3 About the MAML gradients
How to compute the gradient in the MAML flow? Will the embedding network be updated simultaneously? In other words, will the MAML objective influences the embedding network?
3.4 The training details are not clear.
The concrete training setting is not clear. For example, does the method need model pre-train? What is the learning rate, and how to adapt it? For the MAML, we also need the inner-update learning rate. How many epochs does CAML need?
3.5 How about build MAML directly on the embedding space?

---

> ### Author Response · Authors · 2018-11-27
> **Response to reviewer 2**
>
> Thank you for the very detailed and constructive comments.
>
> 1. The motivation
> 1.1 How the metric space is trained?
> The metric space is trained in a pre-training step and it is not updated while training the base-learner. The embeddings obtained from the metric space is different from other popular pre-training techniques, e.g. in LEO the embeddings are pre-trained as a supervised classification task. The pre-trained metric space provides a representation for class dependency as it is trained to provide good separation/clustering from randomly sampled classes. This is in contrast with supervised pre-training which aims to provide discriminative feature representations.
>
> 1.2 “Will it introduce more information w.r.t. only using embedding space to do the classification?”
> The proposed CAML makes use of two views of the data: a global view through the metric space and a local view via the base classifier. The global view of the data, i.e., the embeddings, may not capture all the necessary information for some classification tasks, such as classifying different breeds of dogs which may have similar embeddings. In such cases, the local view of data from the pixel space could help compensate for the lack of information in the global view.
>
> 3.5 “How about build MAML directly on the embedding space?”
> We are aware that meta-learning on the embedding space is a powerful idea, as shown in LEO. We have added an experiment that directly trains maml on the learned 512 dimensional metric space using three fully-connected layers. We were only able to obtain 47.43% on 1-shot tasks and 57.33% on 5-shot tasks. This suggests that applying conditional transformations on the metric space is more effective than directly using the metric space as input.
>
> 2. Novelty.
> The proposed CAML does have close relation to TADAM. However, they have three main differences.
> (i) Different goals: TADAM uses conditional transformation for metric scaling while CAML for developing better gradient-based representations.
> (ii) Task-level vs. example-level representation. TADAM uses task-level representation to modulate the inference from a task perspective, while CAML uses example-level representation to modulate the representation at the content level.
> (iii) The conditional transformation in TADAM is homogeneous in the sense that the conditional information is retrieved from the metric space and also applied to the metric space. However, the proposed CAML uses conditional transformation under the heterogeneous setup where the conditional information is retrieved from the embedding space but applied to a different base learner.
>
> 3. Method details
> 3.1 “Since CBN is example induced, will it prone to overfitting?”
> The metric space (ResNet-12) is pre-trained and not updated while training the base learner. The gradients of the meta learner only affect the base learner and conditional transformation. We choose 30 convolutional channels out of computational considerations, and the skip connection has a bigger impact than the number of conv channels. Using 64 conv channels without the skip connection, we obtain 54.63% on 1-shot and 70.38% on 5-shot.
>
> 3.2 “Is this skip connection very important for this particular model?”
> Yes, the skip connection is very important. The use of skip connections is to improve the gradient flow. MAML unfolds the inner loop into one large graph which may cause gradient issues. Without skip connections, out model obtains 56.07% on 1-shot tasks and 71.26% on 5-shot tasks.
>
> 3.3 “Will the MAML objective influences the embedding network?”
> We would like to clarify that the metric is pre-trained and not updated in MAML updates. We empirically observe that training the metric space and meta-learner end-to-end is overly complex and tend to over-fit.
>
> 3.4 “how many epochs does MAML need?”
> It takes 50,000 episodes to train CAML, and another 30,000 episodes to pre-train the metric space.

---

### Official Review · AnonReviewer1 · 2018-10-29
**Good paper**

**Rating:** 8
**Confidence:** 3

**Review:**

TL;DR. Significant contribution to meta-learning by incorporating latent metrics on labels.

* Summary

The manuscript builds on the observation that using structured information from the labels space improves learning accuracy. The proposed method --CAML-- is an instance of MAML (Finn et al., 2017), where an additional embedding is used to characterize the dissimilarity among labels.

While quite natural, the proposed method is supported by a clever metric learning step. The classes are first represented by centroids and an optimal mapping $\phi$ is then learnt by maximizing a clustering entropy (similarly to what is performed in a K-means-flavored algorithm, though this connection is not made in the manuscript). A conditional batch normalization (Dumoulin et al., 2017) is then used to model how closeness (in the embedding space $f_\phi$) among labels is taken into account at the meta-learning level.

Existing literature is well acknowledged and I find the numerical experiments to be convincing. In my opinion, a clear accept.

* Minor issues

- I would suggest adding a footnote explaining why Table 1 reports confidence intervals and not just standard deviations. How are constructed those intervals?
- Section 3.2 bears ambiguity as the manuscript reads "We first define centroids [...]" depending on $f_\phi$ which is then defined as the argument of the minim of the entropy term. What appears as a circular definition is merely the effect of loose writing yet I am afraid it would confuse readers. I would suggest to rewrite this part, maybe using a pseudo-code to better make the point that $f_\phi$ is learnt.

---

> ### Author Response · Authors · 2018-11-27
> **Response to reviewer 1**
>
> Thank you for your valuable review.
>
> 1. Clarification on the metric learning step
> Thank you for the suggestion. The metric is indeed learned in a K-means-flavored way and we have updated our manuscript to reflect that $\phi$ is learned.
>
> 2. How confidence intervals are constructed?
> We sample 600 evaluation tasks from the meta-test classes and report the confidence intervals across all the evaluation tasks. We have updated our manuscript to reflect this.

---

### Official Review · AnonReviewer3 · 2018-11-02
**An interesting paper with some areas yet to exploit.**

**Rating:** 6
**Confidence:** 3

**Review:**

[Summary]
The paper presents an enhancement to the Model-Agnostic Meta-Learning (MAML) framework to integrate class dependency into the gradient-based meta-learning procedure. Specifically, the class dependency is encoded by embedding the training examples via a clustering network into a metric space where semantic similarity is preserved via affinity under Euclidean distance. Embedding of an example in this space is further employed to modulate (scale and shift) features of the example extracted by the base-learner via a transformation network, and the final prediction is made on top of the modulated features. Experiments on min-ImageNet shows that the proposed approach improves the baseline of MAML.

Pros
- An interesting idea of leveraging class dependency in meta-learning.
- Solid implementation with reasonable technical solutions.

Cons
- Some relevant interesting areas/cases were not exploited/tested.
- Improvement over state-of-the-arts (SOA) is marginal or none.

[Originality]
The paper is motivated by an interesting observation that class dependency in the label space can also provide insights for meta-learning. This seems to be first introduced in the context of  meta-learning.

[Quality]
Overall the paper is well executed in some aspects, including motivation and technical implementation. There are, however, a few areas/cases I would like to see more from it so as to make a stronger case.

In terms of generalization, the proposed enhancement to MAML is claimed to be orthogonal to other SOAs that are also within the framework based on gradient-descent, e.g. LEO. It is not quite clear to me that if the use of class dependency can lead to general benefits to alike methods like LEO, or if it is just a specific case for the MAML baseline. Actually, it would be interesting to see how the proposed class-conditional modulation can help other SOA in table 1. Also, more empirical results from other use cases (e.g., other datasets or problems) also help provide more insights here. These augmentation can better justify the value or significance of this work.

In the specific formulation of the approach in Fig 2, it looks to me that the whole system is a compounded framework that combines two classifiers with one (base-learner) producing base representation, and the second injects side-information (e.g., from class-dependency in this case) to modulates the base representation before the final prediction. I just wonder what would happen if similar process keeps on? E.g., by building the third stage that modulates the features from the previous two? Or what if we swap the roles of base-learner and the embedding from the metric space (i.e., using the base-learner to modulate the embedding)? It looks to me that the feature/embedding from both components (in Fig 5 and 6) are optimized to improve separability. The roles they play in this process are also very interesting to get more elucidation.

Another point worth discussion is that the class dependency currently imposed does not see to include hierarchical structure among classes, i.e., the label space is still flat. It would be great if this can be briefly discussed with respect to the current formulation to better inspire the future work.

[Clarity]
The paper is generally well written and I did not have much difficulty to follow.

[Significance]
While the paper is built on an interesting idea, there are still a few areas for further improvement to justify its significance (the the comments above).

---

> ### Author Response · Authors · 2018-11-27
> **Response to reviewer 3**
>
> Thank you for your constructive review.
>
> 1. Is the use of class dependency general or specific to MAML-based methods?
> (1) The benefits of class dependency is not restricted to MAML-based methods. The goal of class dependency is to provide complementary information to the meta-learner; this is especially important in few-shot learning due to insufficient data.
> (2) Relating to other SOA: (a) TADAM makes use of conditional transformations based on tasks representations for metric scaling; class dependencies can also be incorporated into TADAM with the additional benefit of capturing example-level class relationships. (b) LEO can also make use of the class dependency for improving the conditional generation of model parameters.
>
> 2. The relationship between the metric space and the base-learner.
> The proposed framework captures the dual views of a classification task: a global view that is aware of the relationships among all classes, and a local views of the current N-way K-shot classification task. The metric space, or the global view, is pre-trained in a way that is independent of the current N-way K-shot task; while the base-learner, or the local view, attempts to develop representations for the current classification task alone.
>
> 3. “What would happen if similar process keeps on? E.g., by building the third stage that modulates the features from the previous two?”
> This is a very interesting question. One can build different stages of conditional transformations associated with different granularities of class-dependency. With metric spaces trained to capture different levels of class-dependency, one could modulate the base-learner in a hierarchical manner.
>
> 4. How to make use of hierarchical class structure?
> One can employ a curriculum learning strategy to learn the metric space at different levels of the hierarchy to better train the metric space. As mentioned in 3, the hierarchical class structure can also be used to train different metric spaces and conditionally modulate representations in a hierarchical manner.

---

### Author Response · Authors · 2018-11-27
**Updated version of the paper**

We thank the reviewers for their valuable feedback. The main changes we have made in the manuscript include:

(1) Clarifications on metric learning notations and the fact that the metric space is pre-trained.
(2) Additional discussions about the relationships between the metric space and the base classifier.
(3) Highlight the differences between CAML and TADAM.
(4) Hyperparameters and other small edits.

---

### Meta-Review · Area_Chair1 · 2018-12-14
**Good proposal for incorporating class dependencies in few-shot learning.**

**Confidence:** 4
**Recommendation:** Accept (Poster)

**Metareview:**

The reviewers think that incorporating class conditional dependencies into the metric space of a few-shot learner is a sufficiently good idea to merit acceptance. The performance isn’t necessarily better than the state-of-the-art approaches like LEO, but it is nonetheless competitive. One reviewer suggests incorporating a pre-training strategy to strengthen your results. In terms of experimental details, one reviewer pointed out that the embedding network architecture is quite a bit more powerful than the base learner and would like some additional justification for this. They would also like more detail on the computing the MAML gradients in the context of this method. Beyond this, please ensure that you have incorporated all of the clarifications that were required during the discussion phase.